# *Genopo*: a nanopore sequencing analysis toolkit for portable Android devices

Hiruna Samarakoon [1], Sanoj Punchihewa [1], Anjana Senanayake[1], Jillian M. Hammond [2], Igor Stevanovski[2], James M. Ferguson [2], Roshan Ragel [1], Hasindu Gamaarachchi [2,3,5✉] & Ira W. Deveson [2,4,5✉]

The advent of portable nanopore sequencing devices has enabled DNA and RNA sequencing to be performed in the field or the clinic. However, advances in in situ genomics require parallel development of portable, offline solutions for the computational analysis of sequencing data. Here we introduce *Genopo*, a mobile toolkit for nanopore sequencing analysis. *Genopo* compacts popular bioinformatics tools to an Android application, enabling fully portable computation. To demonstrate its utility for in situ genome analysis, we use *Genopo* to determine the complete genome sequence of the human coronavirus SARS-CoV-2 in nine patient isolates sequenced on a nanopore device, with *Genopo* executing this workflow in less than 30 min per sample on a range of popular smartphones. We further show how *Genopo* can be used to profile DNA methylation in a human genome sample, illustrating a flexible, efficient architecture that is suitable to run many popular bioinformatics tools and accommodate small or large genomes. As the first ever smartphone application for nanopore sequencing analysis, *Genopo* enables the genomics community to harness this cheap, ubiquitous computational resource.

[1] Department of Computer Engineering, University of Peradeniya, Peradeniya, Sri Lanka. [2] Kinghorn Centre for Clinical Genomics, Garvan Institute of Medical Research, Sydney, NSW, Australia. [3] School of Computer Science and Engineering, University of New South Wales, Sydney, NSW, Australia. [4] St Vincent's Clinical School, University of New South Wales, Sydney, NSW, Australia. [5] These authors contributed equally: Hasindu Gamaarachchi, Ira W. Deveson. ✉email: hasindu@garvan.org.au; i.deveson@garvan.org.au

Portable DNA-sequencing devices, such as the MinION device developed by Oxford Nanopore Technologies (ONT)[1], can be used for rapid analysis of genetic material in the field or the clinic. Pioneering studies have demonstrated the utility of in situ DNA and RNA sequencing with a MinION in a variety of contexts, including for Ebola virus surveillance in West Africa[2], profiling microbial communities in the Arctic[3] and genome/transcriptome sequencing on the International Space Station[4]. ONT sequencing has also been widely used to study viral transmission and evolution in real-time during the ongoing 2019–20 coronavirus pandemic (https://artic.network/ncov-2019).

Despite these successes, the full potential of the MinION and related ONT devices for in situ genomics has yet to be realised. This owes partly to the current lack of portable, offline solutions for the analysis of the sequencing data that they generate.

ONT devices measure the displacement of ionic current as a DNA or RNA strand passes through a biological nanopore[1]. The device periodically outputs a group of reads in the form of raw current signals (packed into a .fast5 file) that are subsequently base-called into sequences (a .fastq file). After base-calling, which can be run on a laptop or a portable MinIT device, analysis of the resulting sequence reads is typically performed using dedicated high-end server computers or cloud services (the latter requiring data upload over a high-bandwidth network). This can be an obstacle to ONT field applications. During Ebola surveillance in West Africa, for example, researchers regularly encountered Internet connectivity issues where 3G signals dropped to 2G, massively increasing data upload times[2]. Therefore, while sequencing and base-calling processes are now portable, the computational resources required for downstream bioinformatic analyses, such as sequence alignment and genome assembly, remain prohibitive to in situ genome analysis.

To help address this, we have developed *Genopo*, the first ever mobile toolkit for ONT-sequencing analysis. *Genopo* compacts popular bioinformatics tools to an Android application, suitable for smartphones and tablet computers, thereby enabling fully portable computation. Here, to demonstrate its utility, we use *Genopo* to determine the complete genome sequence of the human coronavirus SARS-CoV-2 in patient isolates sequenced on an ONT device. We then use *Genopo* to profile DNA methylation in a human genome sample, highlighting a flexible architecture that ensures the application is suitable for an array of common genome analyses.

## Results

*Genopo* is an Android application designed to enable fast, portable analysis of ONT-sequencing data on smartphones and tablet computers (Fig. 1). The application can execute a bioinformatics workflow on a nanopore-sequencing dataset copied or downloaded to the device storage (Fig. 1a). We have ported a range of popular tools to *Genopo*. This catalogue currently includes *Minimap2*[5], *Samtools*[6], *Bcftools*[6], *Nanopolish*[7], *f5c*[8], *Bedtools*[9], *Bioawk* and will continue to grow according to the requirements of the community. The user can select to run an individual tool, a combination of tools or an entire pre-built workflow (Fig. 1b). An intuitive graphical user interface within the *Genopo* application allows the user to select their desired tools and configure common usage parameters (Fig. 1c). A terminal environment is also provided within the application, enabling an advanced user to enter command-line arguments. When the user starts the execution, output files are written to the phone storage and the log output is displayed on the application in real-time (Fig. 1d).

To demonstrate the potential utility of *Genopo* for in situ genome analysis, we analysed ONT-sequencing data obtained by targeted amplicon sequencing of the SARS-CoV-2 genome in nine de-identified patient specimens collected at public hospital laboratories in Sydney, Australia. Each isolate was sequenced to a minimum ~200-fold coverage over the ~30 kb SARS-CoV-2 genome (see "Methods" section; Supplementary Table 1). We

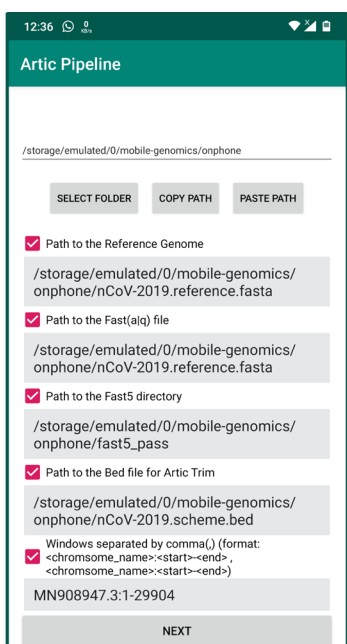
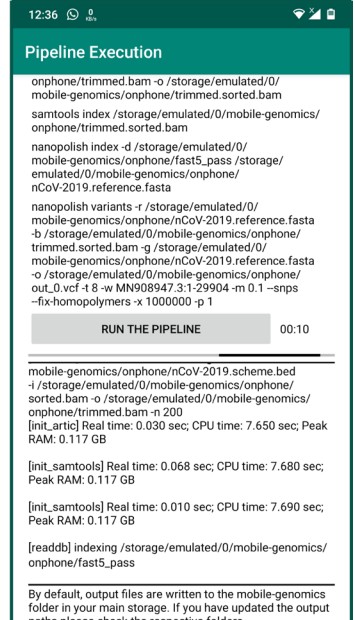

**Fig. 1 Schematic overview of nanopore-sequencing analysis with *Genopo*. a** After sequencing and base-calling, which can be performed on a portable MinIT device (pictured), data are copied to phone storage then analysed with *Genopo*. **b–d** Example screenshots illustrate the selection of tools/workflows (**b**), tool configuration (**c**) and workflow execution (**d**) via the intuitive user interface on *Genopo*.

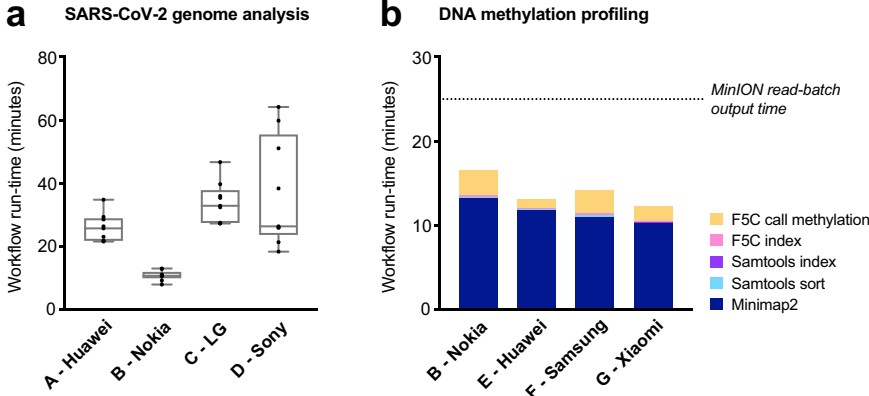

**Fig. 2 *Genopo* run-time performance on various Android smartphones. a** Time taken to complete SARS-CoV-2 genome analysis workflow on ONT libraries from patient isolates ($n = 9$ independent specimens). Box plots show median ± range. **b** Time taken to complete DNA methylation profiling workflow on a single batch of 4000 ONT reads. Individual components of the workflow are represented separately. Estimated time (~25 min) taken per read-batch output for this run is shown for comparison.

used *Genopo* to execute a complete analysis workflow established by the ARTIC network for international coronavirus surveillance (https://artic.network/ncov-2019). This encompassed: (i) read alignment to the SARS-CoV-2 reference genome with *Minimap2*; (ii) compression, sorting, indexing and pre-processing alignments with *Samtools*; (iii) detection of mutations and generation of a consensus genome sequence with *Nanopolish* (Supplementary Note 1.1). The workflow was executed in parallel on four different smartphones, including models from Nokia, Huawei, LG and Sony (Supplementary Table 2).

The entire workflow was completed by *Genopo* in ~27 min per sample, on average, detecting 5–10 mutations and generating a complete consensus genome for SARS-CoV-2 in each patient (Fig. 2a). The vast majority (>98%) of the execution time was consumed by *Nanopolish* variant calling, with the Nokia 6.1 Plus being the fastest model due to its relatively higher RAM (4GB; Supplementary Table 2; Supplementary Data 1). The list of detected mutations and consensus genome sequences were equivalent between smartphone models, and identical to those generated by a best-practice workflow executed in parallel on a high-performance computer (Supplementary Table 1), confirming the integrity of all analyses completed by *Genopo*. Therefore, *Genopo* enables rapid and accurate analysis of coronavirus genome-sequencing data on a generic Android smartphone.

*Genopo* provides an efficient, flexible framework for integrating existing bioinformatics tools to run on Android and is suitable for many popular applications. It not only supports small genomes (e.g., SARS-CoV-2) but can also handle large eukaryotic genomes, including the human reference genome (e.g., GRCh38), using an index partitioning approach established elsewhere[10].

To illustrate this flexibility, we next used *Genopo* to measure DNA methylation signatures in publicly available datasets generated via ONT sequencing of genomic DNA from the popular human reference sample NA12878[11] (see "Methods" section). We obtained datasets containing the output of two MinION flow-cells (Supplementary Table 3) and ran a complete workflow for methylation calling, comprising: (i) read alignment to the full human reference genome with *Minimap2* (GRCh38, eight index partitions); (ii) sorting and indexing of alignments with *Samtools*; (iii) methylation calling using *f5c* (Supplementary Note 1.2).

We analysed the first flow-cell dataset (16,688 reads, 91.15 Mbases) in a single execution on four smartphones, with *Genopo* completing the methylation profiling workflow in ~21 min, on average (Supplementary Table 3). For each execution, a total of

~96,500 CpG sites (30.5%) within analysed reads were identified as being methylated with high probability (log-likelihood ratio > 2.5). Importantly, the proportion and identity of methylated bases was near-identical between different smartphones, and to a best-practice pipeline[7] executed in parallel (99.89%). The majority of the execution time was consumed by read alignment (mean 77.8%) and methylation calling (mean 18.0%; Supplementary Data 2). We noted that the highest and the second-highest peak RAM were recorded for methylation calling with *f5c* (3.4 GB) and read alignment with *Minimap2* (2.2 GB), respectively (Supplementary Data 2). By adjusting the memory governing parameters of each tool, peak RAM can be reduced to support low-end devices (see Advanced Usage Instructions).

An ONT device outputs data as a sequencing run proceeds, theoretically enabling real-time analysis[1]. To test the suitability of *Genopo* for real-time in situ analysis of ONT-sequencing data, we periodically sampled batches of 4000 reads from the second flow-cell dataset (451,359 reads, 3.89 Gbases), thereby mimicking the batch processing behaviour of the ONT base-caller. Batches of reads were assigned to all four smartphones and processed in parallel, using the same methylation-calling workflow as above (Supplementary Note 1.2).

Individual batches of 4000 reads were successfully processed in ~14 min, on average, across the four smartphones (Fig. 2b and Supplementary Table 3). Given a MinION sequencing run-time of ~48 h, the device would produce a batch every ~25 minutes, on average (see "Methods" section). Therefore, any single smartphone would have been adequate to perform real-time DNA methylation profiling for this dataset. These examples serve to illustrate the suitability of *Genopo* for portable, real-time analysis of ONT-sequencing data for a variety of common applications.

## Discussion

The emergence of portable nanopore-sequencing devices has enabled high-throughput DNA and RNA sequencing to be performed in situ without major laboratory infrastructure[1]. The potential utility of ONT devices for field-based genomics applications and rapid point-of-care genetic diagnosis has garnered much excitement, with successful viral surveillance projects (including during the ongoing coronavirus pandemic) providing pertinent examples[2,12,13]. Equally exciting is the possibility that portable and inexpensive sequencing devices will facilitate the democratisation of genome science by reaching otherwise

geographically and/or socioeconomically isolated communities. It is important, therefore, that computational tools developed for the analysis of nanopore-sequencing data are also compatible with these ambitions.

*Genopo* is a flexible toolkit that enables analysis of ONT-sequencing data on Android devices. Here we illustrate the use of *Genopo* for the analysis of sequencing data from SARS-CoV-2 isolates and human genome samples on a variety of popular smartphone models. *Genopo* took ~27 min, on average, to determine the complete SARS-CoV-2 genome sequence in an infected patient. This would represent a relatively small fraction of the typical turnaround time for a typical SARS-CoV-2 nanopore-sequencing experiment, where ~8–12 h is required for sample preparation (reverse-transcription, PCR amplification), library preparation and sequencing. Moreover, when performing DNA methylation profiling in human genome samples, *Genopo* was able to keep pace with the sequencing output of a MinION device. These analyses demonstrate *Genopo*'s suitability for rapid —potentially real-time—ONT-sequencing analysis on a generic Android smartphone.

The smartphone represents a cheap, portable and widely available computational resource that is currently unutilised by the genomics community. *Genopo* allows users to harness this resource for ONT-sequencing analysis. A laptop or MinIT device is currently still required to receive raw data from the sequencer and perform base-calling using ONT's propriety software. However, smartphone-pluggable sequencers, such as the anticipated SmidgION device, will soon bypass this requirement.

We provide *Genopo* as a free, open-source application through the Google Play store. Developers can access the *Genopo* source code and developer's guide at the links provided below (see Code Availabilty). *Genopo* already supports a variety of popular tools and is designed for compatibility with many common bioinformatics analyses, including sequence alignment, variant detection and DNA methylation profiling. For basic usage instructions, please refer to Supplementary Note 2 or the "Help" section within the *Genopo* application.

## Methods

**Genopo implementation**. The *Genopo* Android Application (GUI and the framework) was developed using the Java programming language. Popular bioinformatics tools *Minimap2*[5], *Samtools*[6], *Bcftools*[6], *Nanopolish*[7], *f5c*[8], *Bedtools*[9], *Bioawk* were re-configured and cross-compiled to produce shared libraries (.so files) to run on Android over the ARM processor architecture. The interface between the Android Java application and the native code (compiled shared libraries) was written using the Java Native Interface (JNI). This interface invokes the native functions in the compiled libraries and captures the log output using Unix Pipes. The captured output is displayed on *Genopo* GUI. Android Software Development Kit (SDK) and the Android Native Development Kit (NDK) were used as development tools.

*Genopo* is not just a toolkit but also an open-source framework for integrating existing or future bioinformatics tools into the Android application. Detailed instructions for the reconfiguration and integration of bioinformatics tools into *Genopo*, methods to overcome challenges imposed by the restrictions in the Android Operating system and other Advanced Usage Instructions are provided online.

**Nanopore-sequencing datasets**. Nine de-identified SARS-CoV-2 patient isolates were collected and processed for whole-genome nanopore sequencing for a separate study. RNA extracts from nasal swabs were converted to cDNA, then amplified by multiplexed PCR using 14 × 2.5 kb amplicons, according to a published protocol[14]. Pooled amplicons were multiplexed with native barcoding, according to the standard ONT protocol, pooled, then sequenced together on a single Flongle flow-cell. MinKNOW live base-calling was performed with Guppy 3.2.10 and de-multiplexing was performed with Porechop 0.3.2. Nine de-multiplexed fastq files (one per barcode), the full fastq file (the concatenated version of the nine de-multiplexed fastq files) and the associated fast5 files (raw signal data) were copied on to smartphone devices A, B, C and D (Supplementary Table 2). The SARS-CoV-2 analysis workflow was run independently on each dataset (see below).

**Bioinformatics workflows**. The ARTIC network for viral surveillance (https://artic.network/ncov-2019) has established a standardised bioinformatics pipeline for SARS-CoV-2 genome analysis with ONT-sequencing data. We integrated this workflow into *Genopo* for the analysis of SARS-CoV-2 patient isolates. A detailed description of the workflow is provided in Supplementary Note 1.1. While this workflow is specifically tailored for analysis of SARS-CoV-2, *Genopo* also supports a generic variant calling pipeline that can be used for the analysis of any organism where a reference genome is available.

In addition to variant detection, Genopo also supports a generic pipeline for DNA methylation profiling, which is illustrated via the analysis of publicly available data from ONT sequencing of NA12878. A detailed description of the methylation-calling workflow is provided in Supplementary Note 1.2.

**Calculating base-calling rate**. ONT MinION devices periodically output sequencing data in batches of 4000 reads over the course of a sequencing run (~48 h average duration). Given the total number of reads in a single flow-cell dataset $N$, the number of reads in a base-called batch $B$ and the duration of sequencing run $T$, we can calculate the average time taken by the base-caller to produce a batch (the base-calling rate) $R$ using the following equation:

$$R = (T*B)/N.$$

In our case, $T = 48$ h, $B = 4000$ and $N = 451{,}359$. Therefore, $R = 25.52$ min.

**Reporting summary**. Further information on research design is available in the Nature Research Reporting Summary linked to this article.

## Data availability

All sequencing data used in this study are publicly available. SARS-CoV-2 whole-genome sequencing data have been deposited to the Sequence Read Archive under Bioproject PRJNA651152. Human genome sequencing data from the NA12878 reference sample can be found at the following link: https://github.com/nanopore-wgs-consortium/NA12878/blob/master/Genome.md.

## Code availability

We provide *Genopo* as a free, open-source application through the Google Play store. Developers can access the Genopo source code and developer's guide via the following links: Genopo Play Store: https://play.google.com/store/apps/details?id=com.mobilegenomics.genopo&hl=en; Source code: https://github.com/SanojPunchihewa/f5n and Developers guide: https://mobile-genomics.github.io/genopo/download.html.

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

## Acknowledgements

We thank our colleagues Prof Bill Rawlinson and A/Prof Rowena Bull, who oversaw collection and processing of SARS-CoV-2 patient isolates for a separate study. Thanks also to Chandima Samarasinghe, Harshana Weligampola, Nirodha Suchinthana, Rahal Medawatte and Yasiru Ranasinghe for providing valuable feedback after testing *Genopo*. We acknowledge the following funding sources: MRFF grant APP1173594 (to I.W.D.) and Cancer Institute NSW Early Career Fellowship 2018/ECF013 (to I.W.D.) and philanthropic support from The Kinghorn Foundation (to I.W.D. and H.G.). The contents of the published materials are solely the responsibility of the participating or individual authors, and they do not reflect the views of any funding body listed.

## Author contributions

H.G., R.R. and H.S. conceived the project. H.S., S.P., A.S. and J.F. built and tested the *Genopo* application under supervision from H.G. I.W.D. and H.G. devised benchmarking experiments. J.H., I.S., J.F. and H.G. performed SARS-CoV-2 sequencing and established the analysis pipeline. H.S., H.G. and I.W.D. prepared the manuscript, with support from all co-authors.

## Competing interests

H.G. and J.F. have previously received travel and accommodation expenses to attend ONT conferences. The remaining authors declare no competing interests.
