## [Peer Review File · Communications Biology]

Reviewers' comments:

Reviewer #1 (Remarks to the Author):

In this work, the authors develop a mobile toolkit called Genopo for ONT sequencing analysis on an Android smartphone. Genopo enables fully portable computation as it provides a flexible, efficient framework to run many popular bioinformatics tools (e. g. sequence alignment, generic variant calling, and genome assembly) on Android. This toolkit supports rapid and accurate analysis of sequencing data from SARS-CoV-2 isolates, thereby demonstrating its suitability for in situ genome analysis and show an urgency during the current situation of pandemic. In addition, Genopo is used to profile DNA methylation in a human genome sample at real-time speed. This computational toolkit is very useful and valuable as it allows the genomics community to harness the cheap, portable and widely available computational resource represented by smartphone for nanopore sequencing analysis. I think it's an important work which shows good prospect on many field-based genomics applications and rapid point-of-care genetic diagnosis. It worth publishing ASAP, especially during the current pandemic situation caused by COVID 19.

To make this paper more suitable for publishing, the authors should also address some revisions noted below.

1. Figure 1 (b, c, d), it is difficult to see the information related to selection of tools/workflows clearly. I would suggest the author to either clarify the content that are difficult to be resolved or simplify the demonstration.
2. In the "results" part, paragraph 3, the authors demonstrate that the detected mutations and consensus genome sequences were equivalent between smartphone models, and are identical to those generated by a best-practice workflow executed in a high-performance computer. I would also suggest the authors to be specific with the definition of "a best-practice workflow executed in a high-performance computer" and compare other performance parameters such as the time cost of work completion with the demonstrated workflow.
3. On page 4, the authors demonstrate Genopo can execute data analysis for DNA methylation profiling at real-time speed. This test mimics the batch processing behavior of the ONT base-caller by periodically sampling batches of reads from the existing publicly available dataset. I wonder if Genopo is suitable for other real-time bioinformatic analyses of ONT sequencing data, such as variant detection or genome assembly.
4. And when performing SARS-CoV-2 genome analysis, can Genopo still process sequencing data at a real-time speed?

Reviewer #2 (Remarks to the Author):

Samarakoon et al. present a new toolkit, Genopo, for analyzing nanopore sequencing data on Android smartphones using ARM processors. Genopo is available as an app through the Google Play Store and allows users to customize their workflow using the popular bioinformatics tools for nanopore read mapping, alignment, base-calling etc. The utility of this tool is demonstrated by porting best-practice pipelines for two applications: SARS-CoV-2 genome sequencing and human DNA methylation profiling, on which the tool produces identical or near-identical results.

The tool is easy to use and install and could enable useful field applications, particularly in the times of the pandemic. The evaluation of the tool and description in the text is mostly adequate.

To that end, I am in favor of publication of this manuscript. However, I have a few minor comments and suggestions that should help the authors in revising and further improving the manuscript.

1. The title seems to suggest that Genopo can be used only on smartphones. I guess it could be used on any Android device, including tablets. I was able to install this app on my tablet and try some features and it seemed to work fine. I feel this tool would gain greater traction on tablets (which are now replacing laptops for many users) than smartphones. I suggest they change the title and text to reflect that it can be used on all Android devices unless I am mistaken about this.
2. The authors need to provide version/release numbers for each of the tools ported to Genopo (Minimap2, Samtools etc). Is it easy to update them? Will multiple versions be available on Genopo when tools are updated?
3. Could the authors provide a reference to the best-practice pipeline for DNA methylation pipeline? Why did Genopo not produce identical results to this pipeline (99.89% similarity) and what were the differences and what is causing it?
4. It would be good if authors could provide an estimate of the total time to analyze a SARS-CoV-2 sample, including sequencing and basecalling time. It would put the Genopo computational time under perspective.
5. I believe ARTIC provides version numbers for its protocols. Which version is provided currently in Genopo?
6. Can the authors explain why all devices in Supplementary Table 2 were not used in experiments in Supplementary Table 1 and 4?
7. Can the authors explain the high variance in the Sony device in Fig. 2a and Supplementary Table 1?
8. Since the networking multiple devices feature is not available currently, this should be mentioned clearly in Discussion, or better, skipped entirely.

REVIEWER #1

1.0 In this work, the authors develop a mobile toolkit called Genopo for ONT sequencing analysis on an Android smartphone. Genopo enables fully portable computation as it provides a flexible, efficient framework to run many popular bioinformatics tools (e. g. sequence alignment, generic variant calling, and genome assembly) on Android. This toolkit supports rapid and accurate analysis of sequencing data from SARS-CoV-2 isolates, thereby demonstrating its suitability for in situ genome analysis and show an urgency during the current situation of pandemic. In addition, Genopo is used to profile DNA methylation in a human genome sample at real-time speed. This computational toolkit is very useful and valuable as it allows the genomics community to harness the cheap, portable and widely available computational resource represented by smartphone for nanopore sequencing analysis. I think it's an important work which shows good prospect on many field-based genomics applications and rapid point-of-care genetic diagnosis. It worth publishing ASAP, especially during the current pandemic situation caused by COVID 19.

We thank the reviewer for their careful consideration of the manuscript and constructive feedback. We provide point-by-point responses to each comment from Reviewer #1 below.

1.1. Figure 1 (b, c, d), it is difficult to see the information related to selection of tools/workflows clearly. I would suggest the author to either clarify the content that are difficult to be resolved or simplify the demonstration.

At the reviewer's request, we have reformatted **Figure 1** to make the details more easily readable. High resolution images (PDF format) for both main figures are provided in this resubmission.

1.2. In the "results" part, paragraph 3, the authors demonstrate that the detected mutations and consensus genome sequences were equivalent between smartphone models, and are identical to those generated by a best-practice workflow executed in a high-performance computer. I would also suggest the authors to be specific with the definition of "a best-practice workflow executed in a high-performance computer" and compare other performance parameters such as the time cost of work completion with the demonstrated workflow.

The HPC system we refer to is a Dell PowerEdge R740xd server. We have now added runtime statistics for this HPC system (listed as Device H) to **Supplementary Tables 1 & 3** and provide exact specifications below each table:

"Device H is a high-performance server computer (Dell PowerEdge R740xd). Jobs were run with 384 GB RAM and 32 cores available."

1.3. On page 4, the authors demonstrate Genopo can execute data analysis for DNA methylation profiling at real-time speed. This test mimics the batch processing behavior of the ONT base-caller by periodically sampling batches of reads from the existing publicly available dataset. I wonder if Genopo is suitable for other real-time bioinformatic analyses of ONT sequencing data, such as variant detection or genome assembly.

In the manuscript we show that *Genopo* is suitable to perform DNA methylation profiling in real time. DNA methylation profiling is performed on a per-read basis, with 5-methylcytosine signals detected within individual reads as they are released from the sequencer. *Genopo* is suitable for the real-time execution of any analysis that is performed on a per-read basis. However, processes that utilise information from many independent reads simultaneously, such as genome assembly, can only be performed once the complete sequenced library is available, meaning they cannot be performed in real time. This is a reality of genomics data analysis, not a limitation of *Genopo*.

In many cases, *Genopo* can perform a portion of the overall analysis on-the-fly, and then complete the pipeline's execution once the run has finished, thereby reducing the overall turnaround time. For example, when performing variant detection, *Genopo* can execute read alignment (e.g., *minimap2*) and event alignment (e.g., *nanopolish eventalign*) in real-time, and then complete the process of variant detection (e.g., *nanopolish variant*) at the end of the run.

1.4. And when performing SARS-CoV-2 genome analysis, can Genopo still process sequencing data at a real-time speed?

As explained in the response **1.3**, it is possible to perform part of the SARS-CoV-2 genome analysis pipeline in real time, but not the entire workflow. *Genopo* can perform per-read processes (read alignment, event alignment, primer trimming) as the run proceeds, but can only complete variant detection and generate a consensus genome sequence once all data is available. Again, this is not a limitation of *Genopo* but an inherent characteristic of genomics data.

REVIEWER #2

2.0 Samarakoon et al. present a new toolkit, Genopo, for analyzing nanopore sequencing data on Android smartphones using ARM processors. Genopo is available as an app through the Google Play Store and allows users to customize their workflow using the popular bioinformatics tools for nanopore read mapping, alignment, base-calling etc. The utility of this tool is demonstrated by porting best-practice pipelines for two applications: SARS-CoV-2 genome sequencing and human DNA methylation profiling, on which the tool produces identical or near-identical results.

The tool is easy to use and install and could enable useful field applications, particularly in the times of the pandemic. The evaluation of the tool and description in the text is mostly adequate. To that end, I am in favor of publication of this manuscript. However, I have a few minor comments and suggestions that should help the authors in revising and further improving the manuscript.

We thank the reviewer for their careful consideration of the manuscript and constructive feedback. We provide point-by-point responses to each comment from Reviewer #2 below.

2.1. The title seems to suggest that Genopo can be used only on smartphones. I guess it could be used on any Android device, including tablets. I was able to install this app on my tablet and try some features and it seemed to work fine. I feel this tool would gain greater traction on tablets (which are now replacing laptops for many users) than smartphones. I suggest they change the title and text to reflect that it can be used on all Android devices unless I am mistaken about this.

The reviewer is correct: *Genopo* can be installed and run on any Android device, which includes smartphones and tablet computers. We thank the reviewer for pointing this out and we have decided to change the article's title to:

Genopo: A nanopore sequencing analysis toolkit for portable Android devices.

We have also made several amendments to the manuscript text, in order to highlight the fact that *Genopo* can be run on both smartphones and tablets:

“To help address this, we have developed Genopo, the first ever mobile toolkit for ONT sequencing analysis. Genopo compacts popular bioinformatics tools to an Android application, suitable for smartphones and tablet computers, thereby enabling fully portable computation.” (Intro, para 4)

“Genopo is an Android application designed to enable fast, portable analysis of ONT sequencing data on smartphones and tablet computers.” (Results, para 1)

“Genopo is a flexible toolkit that enables analysis of ONT sequencing data on Android devices.” (Discussion, para 2)

2.2. The authors need to provide version/release numbers for each of the tools ported to Genopo (Minimap2, Samtools etc). Is it easy to update them? Will multiple versions be available on Genopo when tools are updated?

We have added version numbers for all tools to the detailed bioinformatics workflow descriptions in **Supplementary Note 1 & 2**. These indicate the software versions used within *Genopo* during the preparation of the present manuscript, which are not necessarily the latest versions currently hosted on *Genopo*.

The application is designed so that hosted tools can be easily updated and this will be actively maintained by the developers to ensure up-to-date versions are available. A single stable version of each tool will be hosted, and multiple versions will not be simultaneously supported. However, we use github releases (<https://github.com/SanojPunchihewa/f5n/releases/>) to host all the release versions (latest version and all past versions) of the Android Application Packages (APK). If a user requires an older software version they can download and install by following the step-by-step guide provided at (<https://mobile-genomics.github.io/genopo/download.html>). If the user wants a particular version of the software that is not available in any of the GitHub releases, they can build *Genopo* from the source code with the particular versions of the software they need. The step-by-step guide for this building is provided at (<https://mobile-genomics.github.io/genopo/support.html>).

2.3. Could the authors provide a reference to the best-practice pipeline for DNA methylation pipeline? Why did Genopo not produce identical results to this pipeline (99.89% similarity) and what were the differences and what is causing it?

The best practises pipeline refers to the steps presented in:

https://nanopolish.readthedocs.io/en/latest/quickstart_call_methylation.html

We have added the relevant citation in the text:

Simpson, Jared T., et al. "Detecting DNA cytosine methylation using nanopore sequencing." *Nature Methods* (2017).

This minor discrepancy results from the use of index partitioning by *Genopo*. On the HPC system (Device H – see **1.2**) *minimap2* read alignment is executed on the whole reference genome index, held in the RAM, whereas *Genopo* partitions the index (because insufficient memory is available on a smartphone device to hold the entire index in the RAM). When index partitioning is similarly used on the HPC, the results are identical.

Variation of results between whole index and peartitioned index approaches is mainly associated with low-complexity regions. *Minimap2* may generate suboptimal alignments in such regions and the generated results may slightly differ between a whole index and a partitioned index. This only affects a tiny fraction of reads, which tend to have low mapping quality.

A thorough analysis of these differences are presented in the following article, which we have cited in the present manuscript:

Gamaarachchi, H., Parameswaran, S. & Smith, M.A. Featherweight long read alignment using partitioned reference indexes. *Scientific Reports* 9, 4318 (2019).

2.4. It would be good if authors could provide an estimate of the total time to analyze a SARS-CoV-2 sample, including sequencing and basecalling time. It would put the Genopo computational time under perspective.

The workflow will vary somewhat from lab to lab, depending on the protocols adopted, but for the standard ARTIC protocol ~4-5 hours is required for sample processing (reverse transcription and PCR amplification), ~2-3 hours for library preparation and ~2-3 hours for sequencing.

At the reviewers suggestion, we have added some discussion of the turnaround time to the manuscript discussion:

"Genopo took ~27 minutes, on average, to determine the complete SARS-CoV-2 genome sequence in an infected patient. This would represent a relatively small fraction of the typical turnaround-time for a typical SARS-CoV-2 nanopore sequencing experiment, where ~8-12 hours is required for sample preparation (reverse-transcription, PCR amplification), library preparation and sequencing." (Discussion, para 2)

2.5. I believe ARTIC provides version numbers for its protocols. Which version is provided currently in Genopo?

Genopo currently hosts ARTIC version 1.0.8, which has some minor differences in software versioning, compared to the latest ARTIC pipeline (1.1.3):

Software	ARTIC on Genopo (1.0.8)	Latest ARTIC (1.1.3)
nanopolish	v.0.11.3	0.13.2
minimap2	2.17-r974	2.17
samtools	1.10	1.9
bcftools	1.10.2	1.9

While these updates are relatively minor, we will ensure that the latest ARTIC version, including all software updates, is hosted on *Genopo* at the time of publication for this article.

2.6. Can the authors explain why all devices in Supplementary Table 2 were not used in experiments in Supplementary Table 1 and 4?

This was simply due to practical reasons. All experiments were performed on smartphones borrowed from willing undergraduate students at the University of Peradeniya. The set of devices used was dictated by availability at the time a given benchmarking experiment was being performed and therefore differs between the NA12878 and SARS-CoV-2 experiments. This shows that *Genopo* can be used on a wide range of smartphone models.

2.7. Can the authors explain the high variance in the Sony device in Fig. 2a and Supplementary Table 1?

Unfortunately, we no longer have access to this device in order to investigate the specific cause of the high variance in run-time. However, we suspect that RAM limitations, competing background applications and the behaviour of the swap space (the space allocated on the disk / flash memory to facilitate larger virtual memory than the physical memory) all interact to cause this issue.

The device has limited RAM – 3GB in total – which is shared by the operating system, vendor applications and user applications. RAM availability is critical for run-time. For example, if 1.2 GB of free RAM (of 3 GB) was available and if *nanopolish* consumed above 1.2 GB for particular genomic windows in certain datasets, swapping of memory pages will occur between the physical RAM and swap space. Such swapping can cause a considerable performance drop as the flash memory is very slow compared to RAM. Memory swapping can also be caused by a sudden increase of memory consumption by a background application, because this reduces the amount of free RAM available for *Genopo*.

The Sony device had relatively small RAM compared to other devices (3GB), making it more likely to encounter competition for RAM. Sony smartphones also tend to come with a number for pre-installed vendor applications which may intermittently consume RAM, thereby causing conflicts during some pipeline executions but not others.

2.8. Since the networking multiple devices feature is not available currently, this should be mentioned clearly in Discussion, or better, skipped entirely.

We take the reviewer’s point: this is future work that has not yet been implemented. Therefore, we have removed the following sentence from the discussion:

~~“Additionally, while we have only shown single-phone applications here, multiple phones may be networked to handle large data volumes with *Genopo*.”~~